



# Classification of Sea-Ice Concentration from Ship-Board S-Band Radar Images Using Open-Source Machine Learning Tools

Elizabeth Westbrook[1], Peter Gaube[2,4], Emmett Culhane[3], Frederick Bingham[1], Astrid Pacini[2], Carlyn Schmidgall[2,4], Julian Schanze[5], and Kyla Drushka[2]

[1]University of North Carolina - Wilmington North Carolina, United States of America
[2]Applied Physics Laboratory - University of Washington - Seattle Washington, United States of America
[2]Applied Physics Laboratory - University of Washington - Seattle Washington, United States of America
[3]Woods Hole Oceanographic Institution - Woods Hole Massachusetts, United States of America
[2]Applied Physics Laboratory - University of Washington - Seattle Washington, United States of America
[4]School of Oceanography - University of Washington - Seattle Washington, United States of America
[2]Applied Physics Laboratory - University of Washington - Seattle Washington, United States of America
[5]Earth and Space Research - Seattle Washington, United States of America

**Correspondence:** Peter Gaube (pgaube@uw.edu)

**Abstract.** The 2022 NASA Salinity and Stratification at the Sea Ice Edge (SASSIE) expedition measured ocean surface properties and air-sea exchange approximately 400 km north of Alaska and in the Beaufort Sea. The survey lasted 20 days, during which time screen captures from the shipboard S-band radar were collected. Our goal was to analyze these images to determine when the ship was approaching ice, in the ice, or in open water. Here we report on the development of a machine learning
5    method built on the PyTorch software packages to classify the amount of sea ice observed in individual radar images on a scale from L0-L3, with L0 indicating open water and L3 assigned to images taken when the ship was navigating through thick sea ice in the marginal ice zone. The method described here is directly applicable to any radar images of sea ice and allows for the classification and validation of sea ice presence or absence.

## 1 Introduction

10    In the Arctic, the advance of the autumn sea ice has become slower and occurs later in the season, whereas the retreat of the summer ice has become faster and occurs earlier in the season, both as a direct result of climate change (Stroeve and Notz, 2018). As more of the ocean surface is exposed to the atmosphere during longer periods of open water, air-sea heat and momentum fluxes, the freshwater cycle, surface albedo feedback, primary production, and regional and global climate, as well as human and ecological health are changing (Lannuzel et al., 2020). The 2022 NASA Salinity and Stratification at the

15    Sea Ice Edge (SASSIE) mission aimed to study the role that surface salinity and stratification have on controlling the freezing of sea ice. The SASSIE expedition took place in the marginal ice zone during fall 2022 in the western Beaufort Sea around 72°–73.5°N, 154°–145°W (Drushka et al., 2024). This area was generally covered with sea ice year-round until the late 2000s, when the marginal ice zone moved northward during the summer melt season. Drastic changes in sea ice cover are occurring throughout the Arctic and direct observations of sea ice are needed to help understand and document them.



Direct observation of sea ice concentration is important because passive microwave satellite measurements struggle to quantify SIC during the melt season (e.g., Kern et al., 2020). Ship-based S-band radar has been shown to be useful in the identification of sea ice, although other wavelengths can prove to be better for sea-ice characterizations (Haykin et al., 1985). Lund et al. (2018) used a shipboard X-band marine navigation radar to detect and track sea ice. Ku-band radar was used to characterize the types of sea ice observed, be it first-year ice, multilayer ice, icebergs, or the shadows cast by icebergs (Orlando et al., 1990). However, classifying individual sea-ice radar images is labor intensive. In this study we develop a machine-learning method to characterize sea ice concentration from images captured by a shipboard S-band radar, which would otherwise be used for navigational purposes only.

The classification of sea ice types from S-band radar images lends itself well to the use of machine learning models because the images are invariant to rotation. The ubiquity of machine-learning classification methods in Earth Science has resulted in open-source tools that can be implemented on diverse datasets without the need to develop complex software required to build such models. In this project we use the open-source PyTorch machine learning library *https://pytorch.org/*.

## 2 Methods

### 2.1 S-Band Radar Data Collection

The SASSIE expedition took place aboard the *R/V Woldstad* which had a bow-mounted Furuno 2137S-BB S-band marine navigation radar. The primary purpose of the S-band radar is to provide the ship's captain with imagery of the surrounding ice field to assist in safely navigating through the ice. We recorded images of the radar by splitting the signal betwen the display and a screen capture device (Epiphan AV.io HD). This video feed was recorded using a command line video conversion program called "ffmpeg" available at *ffmpeg.org* that saved screenshots as jpeg files typically every 60 s while the ship was in and around the ice. During autonomous vehicle deployments on the ice, images were acquired every 10 s to map the evolution of the ice at higher resolution (Drushka et al., 2024). While the ship was in ice-free water for more than 1 day, S-band acquisition was paused.

### 2.1.1 S-Band Radar Data Processing S-Band Radar Data Processing

Details on how the screen capture device was set up and the steps used to convert the images (Fig. 1a) into geolocated bit maps (Fig. 1b) are described in detail in Drushka et al. (2024) and summarized below. The metadata was extracted from the images using MATLAB's computer vision toolbox. Metadata include the spatial range of the image, which varied in size from 1.4 km to 22 km. The pixels outside of this range were removed from each image, and pixels within the range were georectified and stored as Geo-TIFF images. The pixels were classified according to their red, green and blue (RGB) values. Color values consistent with yellow to red color ranges were classified as "sea ice", and those consistent with blue and green colors were classified as "no data". "Sea clutter" is noise near the center of the radar image that results from surface waves reflecting the radar signal (e.g., prominent yellow-red spot in the center of Fig. 1a). Sea clutter appears in every radar image but its extent and





intensity varies depending on factors including the sea state and the range setting of the radar. Visual inspection showed that sea clutter typically does not appear at a distance greater than 5% of the image's range. In regions of heavy ice cover, sea clutter overlaps with sea ice. As a result, sea clutter could not be removed with a simple mask and was instead removed as follows. RGB values within 5% of the image range were summed as a function of distance from the center of each image ($\Sigma RGB$); sea clutter appears as a peak in $\Sigma RGB$. The edge of the sea clutter was defined as the distance from the center at which sea clutter could no longer be detected, which was identified as the distance at which $\Sigma RGB$ dropped below 5% of its peak value. If no values of $\Sigma RGB$ were below this threshold, the threshold was increased iteratively until $\Sigma RGB$ dropped below the threshold. Pixels at smaller distances from center were flagged as sea clutter. The flagged images were stored as Level-4 (L4) Geo-TIFF images (e.g., Fig. 1b). In total, more than 24,000 L4 files were produced from the processing of the S-band images.

## 2.2 Classification of Sea-Ice Concentration Using L4 Geo-TIFF Images

The classification of sea ice along the ship track of the SASSIE expedition was first attempted as a simple ratio of black to white pixels. This did not work for the following reasons:

### 2.2.1 Noise in the L4 Images from Unidentified Sea Clutter

Due to turbulent water around the ship and noise from the S-band radar, it is common to see what looks to be small ice floes around the ship that are not actually ice. These range in intensity from a few splotches (Fig 2a) to a full ring of white around the ship (Fig 2b). This occurred mostly in cases where the ship was in open water, but also happened sometimes when the ship was in ice (Fig 2c). These are likely sea clutter that was not identified by the algorithm described above. Since this sea clutter is unidentified, it is not possible to distinguish from genuine ice floes. However, the appearance of unidentified sea clutter and its typical proximity to the ship itself make it possible to recognize with a ML model.

### 2.2.2 Distortion of Ice Due to the Leading Edge Detection

When the ship approaches a patch of sea ice, the amount of ice in the floe is underestimated due to the radar's detection of the leading edge of the ice. This is revealed in two radar captures taken 3 minutes apart as the ship enters a dense patch of ice (Fig. 3). This leading edge effect results in a underestimation of ice concentration in close proximity to the ship when simple ratios of black and white pixels are used to analyze these images. In the machine learning model described below we are able to account for this distortion by including numerous examples in the training data.

### 2.2.3 Radar Setting Changes Impact How Sea-Ice is Observed

Since the primary purpose of the S-band radar was to allow for safe navigation of the *R/V Woldstad* through the ice field, radar settings were changed throughout the campaign depending on ice conditions, weather conditions, and captain preference. Ice fields on the radar look different depending on these settings, specifically the set range of radar detection. Two cases where the ship is in heavy ice with different ranges on the radar are shown in Fig 4. When the radar is 'zoomed-out' to show a large area,





ice near the ship appears much smaller and ice around the edges of the frame is not picked up on the radar (Fig. 4a) whereas when zoomed in, more ice is detected near the ship (Fig. 4b). The method we propose below is insensitive to this effect as the model is trained on images at all zoom levels.

## 2.3 The Machine Learning Model

These issues with using S-band radar to classify ice led us to abandon our simple ratio estimation and develop a machine learning model to classify the images. To this end, a subset of about 1,100 of the 24,000 L4 Geo-TIFFS were chosen in non-sequential order to remove near-duplicates in order to develop a hand-curated machine learning training data set. The training images were labeled using a simple 4-level classification with ice concentration ranging from a zero (C0) to three (C3). We chose to use ice levels versus directly estiuamte ice coverage because our interest is to identify if the ship was in ice, around ice,

or no ice. This 4-level classification has a values of C0 when there is no ice at all in the frame of the radar image and the ship is in truly open water (Fig 5a). Noteworthy variation in the L0 images was observed due to different amounts of "noise" present on the radar due to remote instruments (Wave Gliders, SWIFT buoys, etc.; see Drushka et al. (2024)). The C1 ice concentration ranges from having just one or two ice floes in the frame of the image to ice fields with a density less than 10 percent (Fig 5b). For radar captures where the ship was in low to medium concentrations of ice, or there is high concentration of ice in the frame

but the ship is not passing through it, a C2 classification was used (Fig 5c). Finally, a C3 classification was used to indicate that the ship is passing through sea ice with concentration in excess of about 10% (Fig 5d). Higher sea ice concentrations were not encountered because the ship does not have ice breaking capabilities.

The machine learning (ML) model we used for image classification is an the off-the-shelf implementation of the VGG-19 network in PyTorch (Simonyan and Zisserman, 2014). The original VGG-19 is designed for the ImageNET benchmark task,

and thus we reduce the final fully connected layer from 100 to 4 dimensions, and leverage transfer-learning by initializing the model with pre-trained weights (Yosinski et al., 2014). Predictions are generated using the softmax activation function on the final fully connected layer (Bridle, 1990). VGG-19 is a complex and computationally expensive architecture, and so risks overfitting and can require the use of multi-core GPUs on large datasets (Huang et al., 2017). Here, we are able to run the model to convergence on a local CPU in less than an hour, as the single-channel image dataset is just 26MB on

disk. We prevent overfitting by regularizing the model through the use of dropout and data augmentation. Dropout in a CNN (Convolutional Neural Network) is a regularization technique that randomly deactivates a fraction of neurons during training to prevent overfitting and improve the network's ability to generalize to new data (Srivastava et al., 2014). Our best performing model used dropout probabilities of 0.6 on the two fully connected layers, such that 60% of the neurons in these layers are randomly set to zero for each training example. In addition to dropout, we regularized the model using a data augmentation

protocol that took advantage of the rotational invariance of the images. The S-band radar observes a circular region around the vessel, and thus the assigned class should be invariant to any rotational changes to the input. Here, we used an augmentation protocol that rotated 75% of the images vertically and/or horizontally by up to 50%, such that just 12.5% of input images are fed to the model in their original orientation (as in Bridle, 1990). The model was trained for 15 epochs using a batch size of 64, and a train/test split of 85/15 and 5-fold cross-validation. The model achieved 91% accuracy on the test data, with class-specific





performance shown in Table 1 and confusion matrix in Figure 6. The model shows high performance on all 4 classes, and does not misclassify any of the testing data by more than one ice level (ex. C0 are only misclassified as C1).

     The ML model is well suited to use images that contain random anomalies like those from sea clutter in the S-Band radar images, and CNNs are particularly well-suited for tasks like this where visual patterns are perceptible to humans but difficult to formalize into specific rules. Since CNNs can learn these insights directly from labeled data (e.g., human-labeled ice levels),

the model can be trained without requiring hand-crafted rules or filters. This is ideal in situations where human experts can label examples but cannot precisely specify what distinguishes one level of ice from another in terms of pixel properties. CNNs bridge this gap by mapping visual patterns to labels directly, learning filters that capture various levels of detail—from edges and textures to larger structural patterns—enabling the network to distinguish between visual nuances like the difference between low-ice concentrations and sea-clutter, that resist more simplistic rules-based methods.

## 3   Validation


### 3.1   Photo and Logbook Validation

Throughout the SASSIE expedition, notes documenting the state of the ice were recorded in the ship's log and geolocated photos were taken from the deck of the *R/V Woldstad* while the ship was in ice. To validate the predictions of our model we compiled a 'ground-truth' dataset from the images and logbook entries. Figure 7a-c provides examples of photos from the cruise

that are classified as C1, C2, and C3, respectively. Observations from the photos and ship's log were tabulated and compared with the moving hourly average from the S-band ice classification product, which is shown in Fig. 8 below revealing general agreement between the model classification (black curve) and the ship-board photos (blue points) and ship's log (red points). In general the model estimates agree with both the photo and log book observations, however the point-wise comparison lacks the granular detail required to compare the high-frequency S-band radar observations to validation data and the time stamps

of the log book entries were often rounded to the nearest hour, resulting in some mismatch between the log book observations and our S-band based characterization.

### 3.2   Comparisons to Regional Ice Coverage Maps

To assess the validity of the model predictions of sea ice classification we compare our estimates to those of a derived daily gridded sea ice product generated from operational sea ice maps made by the National Weather Service Alaska Sea Ice Pro-

gram (grASIP). The details of how the product is produced are given in Pacini et al. (2024). Following the shiptrack, general agreement is found between sea ice concentration estimated from the model developed here and that of the grASIP (Fig. 9). Regression of our classification to the daily grASIP sea ice concentrations yields an $r^2 = 0.78$.





## 3.3 Comparison to Temperature and Salinity

The S-band ice products are compared with temperature and salinity measurements from the shipboard thermosalinograph
(Drushka (2024)) and the 'salinity snake' which recorded data at the sea surface throughout the field campaign (Schanze (2024)). Time series of hourly moving S-band product average together with temperature and salinity measurements were constructed and indicate that generally the warmest temperatures were observed when the ship was in open water and the coolest when the ship was near sea ice with generally good agreement between the ice classification and the temperature (Fig. 10).

Visualizing both the temperature and salinity as a function of sea-ice classification (Fig. 11) we can observe that generally C2 or C3 classifications are numerous in fresh and cool water while C0 and C1 radar captures occurred in generally warmer and saltier water. The highest number of C3 classifications (our highest ice concentration) occurred in water cooler than 0 degrees C and fresher than 24 PSU suggests that our automated algorithm is indeed detecting sea ice and not just sea clutter.

## 4 Conclusions

We employed an open-source machine learning model to classify sea-ice concentration trained on a subset of S-band radar images that were classified as C0 (no ice), C1 (some ice), C2 (more ice and near the ice edge), and C3 (in the ice). Our model was successful at predicting a subset of training data between 84% and 95% of the time with the highest skill associated with the C3 label. Using this model we classified over 24,000 images and compared these classifications to both photos taken from the ship as well as the Alaska Sea Ice Program gridded ice maps and shipboard measurements of temperature and salinity. This
simple model yielded predictions that fit into the temperature and salinity expected for differing ice coverage. It is important to note that the middle two classifications (C1 and C2) were difficult to define in the training data and also yielded complex relationships with temperature and salinity suggesting that our methods are best suited for determining if there is ice or not, versus how much ice is really there. Overall, this method provides an example of using existing machine learning methods to create value-added data products from what generally is considered navigation radar and not traditionally used to characterize
sea ice.

## 5 Interactive computing environment

*Code availability.* All the code used to train and run the ML model are available on the project repository https://github.com/NASA-SASSIE/seaice.



*Data availability.* The L4 Geo-TIFFS are available at NASA PO.DAAC, doi:10.5067/SASSIE-SBAND4. Shipboard temperature and salin-
ity data are avalable at doi:10.5067/SASSIE-TSG2. Predictions from the ML model developed here are available also on the PO.DAAC,
doi:10.5067/SASSIE-SBAND-ML.

*Author contributions.* EW, PG, EC, FB, CS, and KD contributed equally to the overall design of the project. EW and EC led the ML model
development. KD processed the raw images into L4 Geo-TIFFS and ship-board TSG data. JS processed the salinity snake data. AP created
the gridded ASIP dataset, in collaboration with the Naitonal Weather Service. PG and EW led the writing of the manuscript.

*Competing interests.* The authors declare that they have no conflict of interest

*Acknowledgements.* The authors thank all the participants that made the SASSIE program possible including support from the NASA
Physical Oceanography Program. We thank Nadya Vinogradova-Shiffer of NASA for her support throughout the SASSIE program. We are
grateful to the captains and crew of *R/V Woldstad* and to Support Vessels of Alaska Inc.





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



(a)  (b)

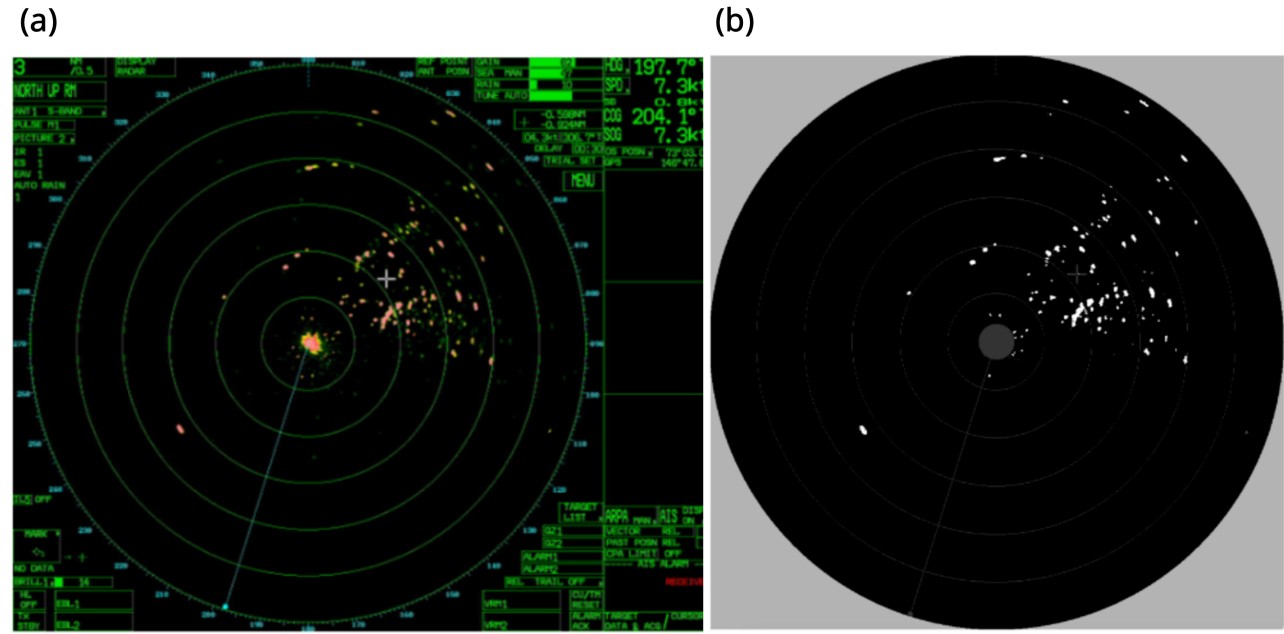

**Figure 1.** (a) Screen capture from the Furuno S-band radar display and (b) the accompanying L4 geolocated image used in this analysis



(a)          (b)          (c)

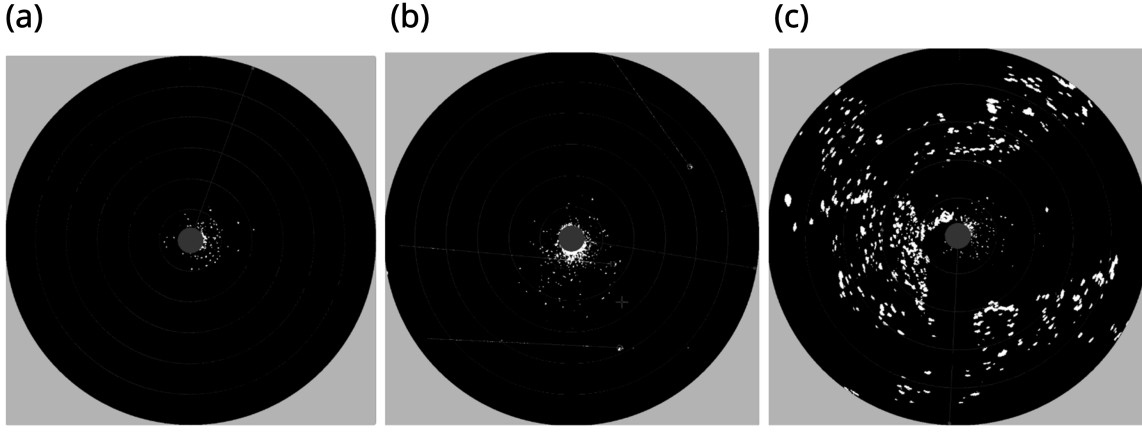

**Figure 2.** (a) Example L4 Geo-TIFF image with minor unidientified sea clutter, (b) major unidientified sea clutter, and (c) unidientified sea clutter while in the ice.



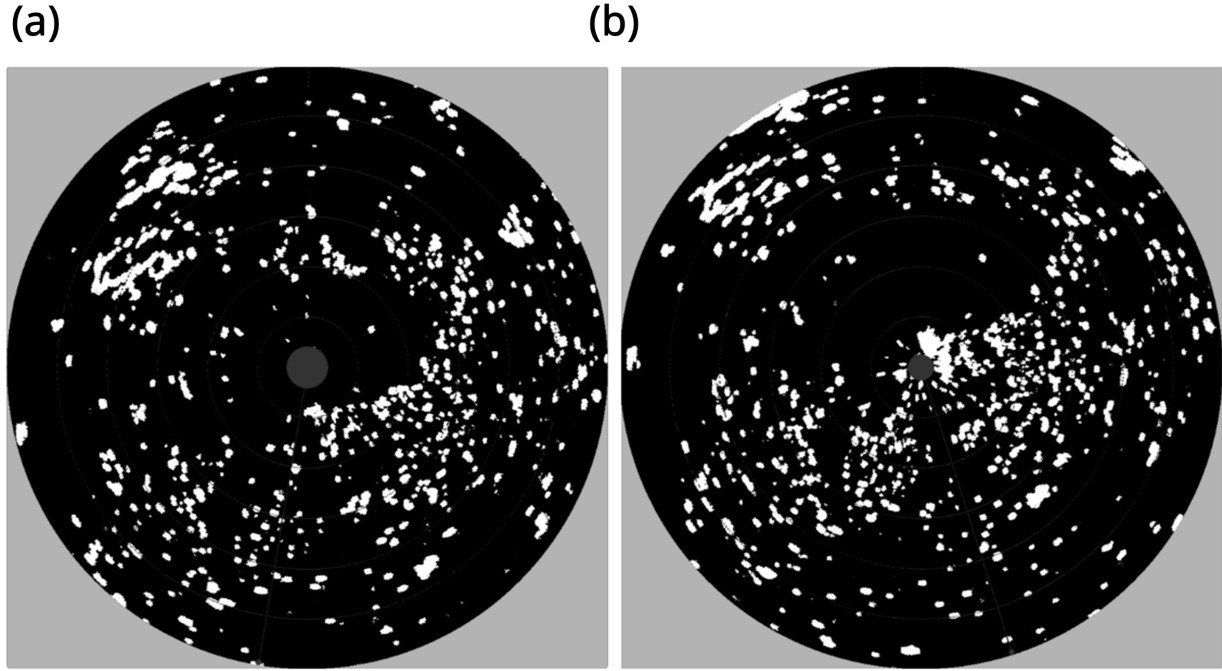

**Figure 3.** Example L4 Geo-TIFF image from September 9 2022 at (a) 17:51 and (b) 17:54 showing the effect of returns off of the leading edge of the ice pack (a) and returns off of ice not visible when first approaching the dense patch of ice (b). Ship is moving from the northwest to the southeast.



(a)                                                  (b)

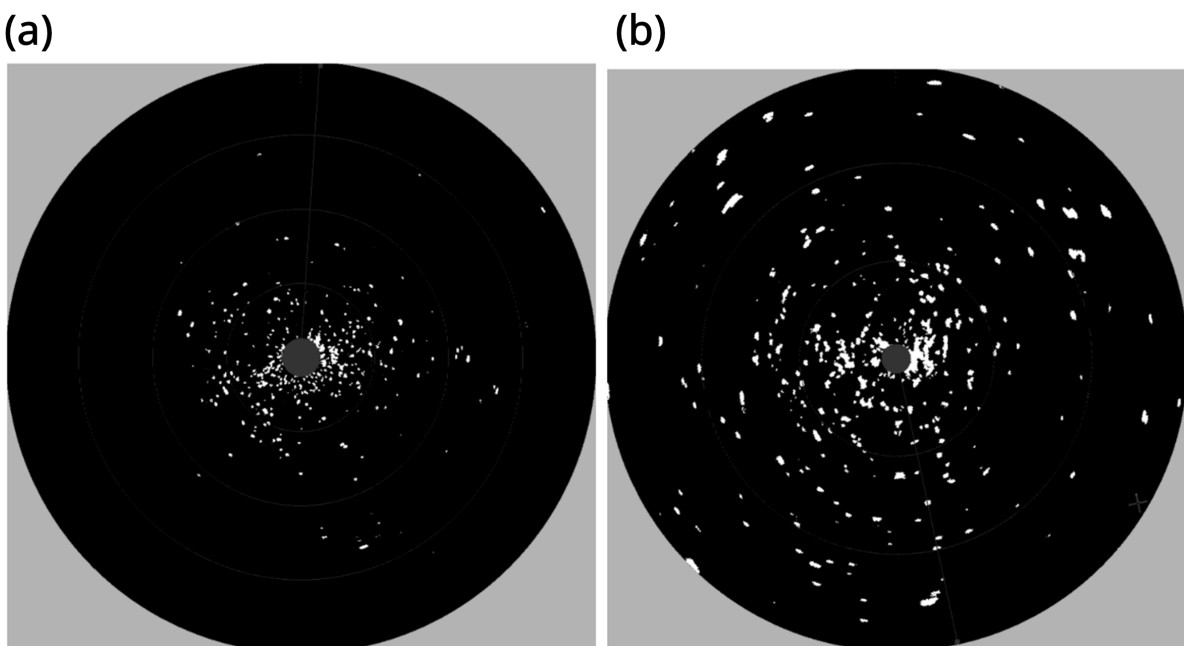

**Figure 4.** Example L4 Geo-TIFF images two minutes apart showing how changes in the radar setting by the ship's captain, in this case changes in radar range with (a) being zoomed out (larger range) in compared to (b), affect the detection of sea ice.





(a)

(b)

(c)

(d)

**Figure 5.** L4 Geo-TIFF image taken from the training dataset showing three examples each of (a) C0, (b) C1, (c) C2, and (d) C3 sea ice classifications.

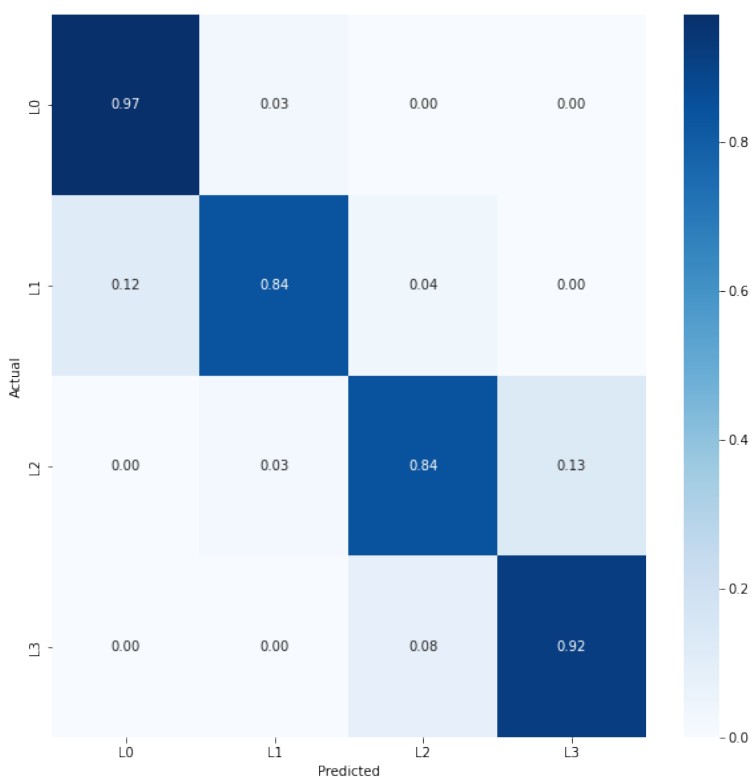

**Figure 6.** Confusion matrix of the predictive performance of the final CNN model used in the analysis. The class accuracy is shown along the diagonal. Notably, no examples in the testing data have been misclassified to an ice level that is more than 1 step away. For example, no L0 are classified into C2 or C3, no C2 are classified as C0, etc



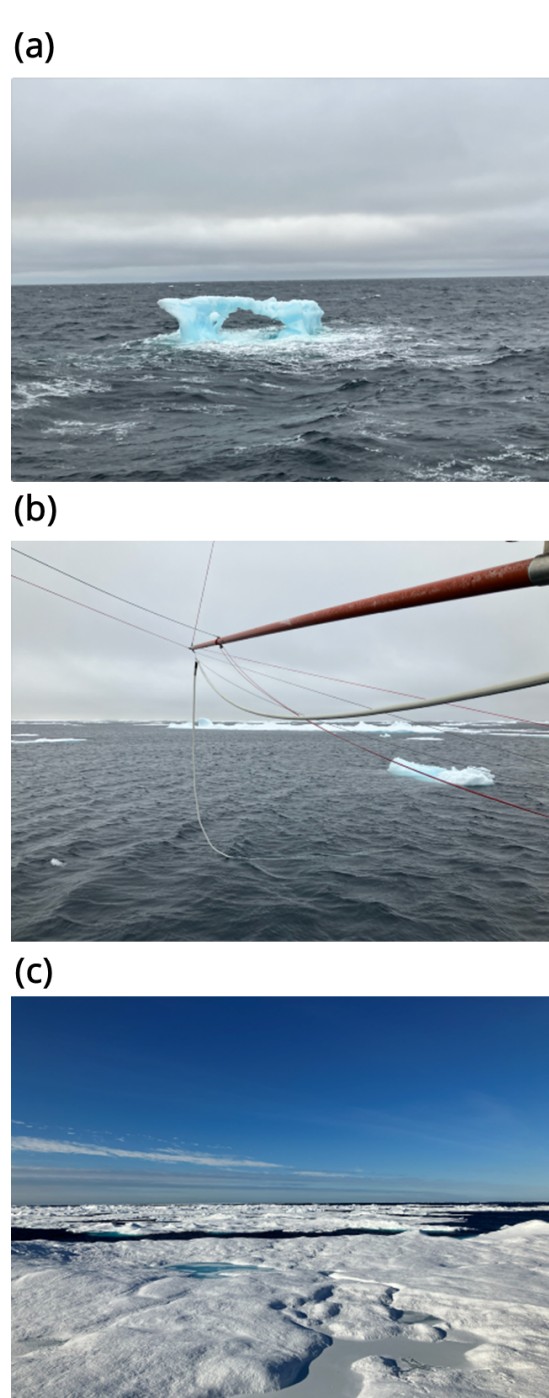

**Figure 7.** Example collocated photographs used for validating the model results showing (a) C1, (b), C2, and (c) C3 sea ice classifications.



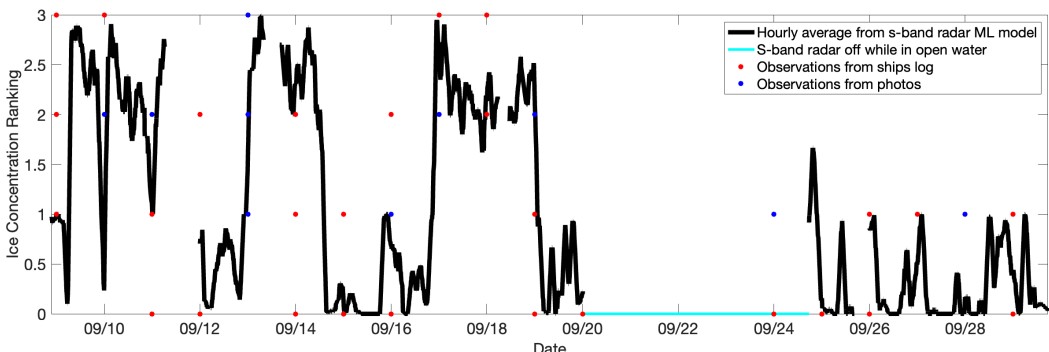

**Figure 8.** Hourly average ice classification (black line) plotted with ice classifications (0-3) assigned to times based on photos from the cruise and notes from the ship's log. Some of the time stamps on notes from the ship's log may be slightly off due to rounding of times when recording.

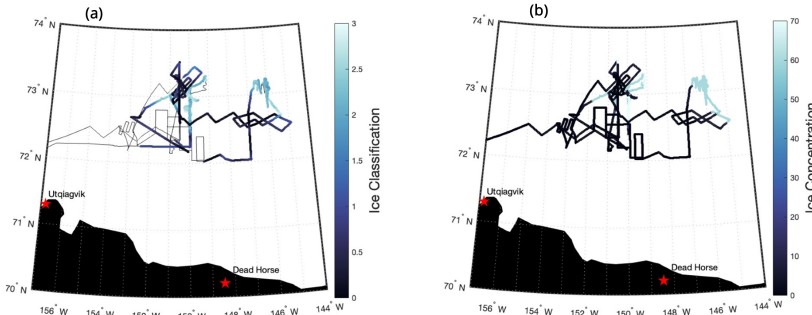

**Figure 9.** Maps of the track of the *R/V Woldstad* during the SASSIE campaign colored by hourly average ice classification from the S-band radar model estimates (a) and the ice concentration from grASIP (b). Grey ship track to the west in (a) are when the S-band radar was turned off.



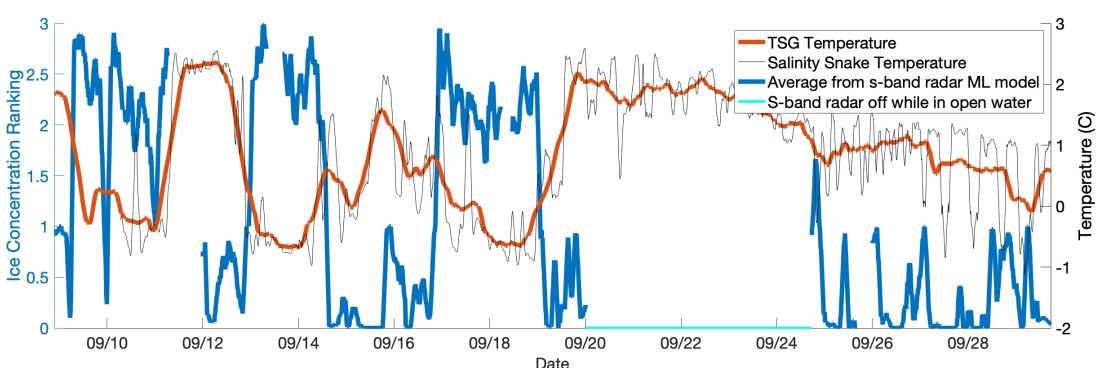

**Figure 10.** Temperature recorded by the shipboard TSG and Salinity Snake throughout the SASSIE field campaign.

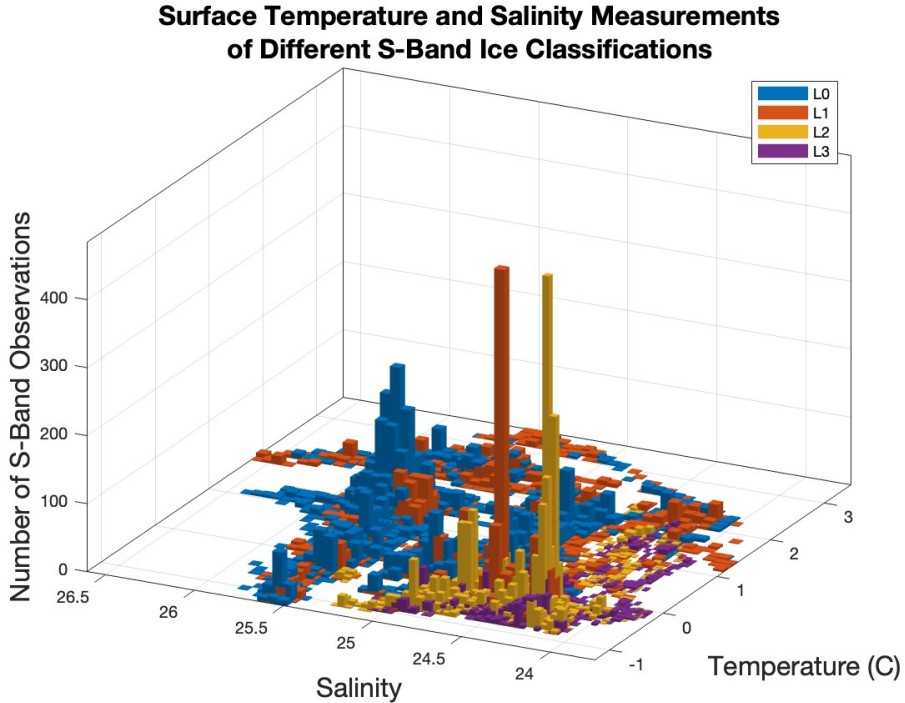

**Figure 11.** The number of measurements recorded from each ice classification in temperature and salinity bins as a 2D histogram. Bar color indicates sea ice classification with the z-axis indicating the number of observations at a given temperature and salinity bin.





**Table 1.** Table showing the model accuracy at predicting a subset of the training data withheld for validation.

| Classification | Accuracy (%) |
| --- | --- |
| C0 | 97 |
| C1 | 92 |
| C2 | 84 |
| C3 | 84 |