# Peer review of "Classification of Sea-Ice Concentration from Ship-Board S-Band Radar Images Using Open-Source Machine Learning Tools"

_EGUsphere, 2025_

## Referee Comment (RC2)

Westbrook et al. (2025) propose a machine learning-based approach for classifying sea ice concentration using ship-board S-band radar imagery. This is a novel contribution with substantial scientific merit. The study is methodologically sound, particularly due to its use of multiple validation datasets, which significantly enhances its robustness.

However, I recommend **major revisions** before the manuscript is ready for publication. One of the key shortcomings is the **absence of a dedicated discussion section**. The current results section does not engage in analytical interpretation, and the conclusion is overly brief, leaving the manuscript feeling incomplete. More critical reflection on the implications, limitations, and broader context of the findings is necessary to fully realize the potential of this work.

Also, The quality of figures need to be improved before publication. The captions need to have more information.

Some general comments that need to be addressed:

Abstract: I feel the abstract needs to establish the uniqueness of the research as well as the ending sentence should put the applicability of this work in broader context.

Introduction:
The introduction is too short and needs more details to set the context and uniqueness of the study. Particularly, focused should be given on why classification of S-band radar and any past studies that has looked into this? Also, how is this helpful from a satellite remote sensing context, is this relevant for upcoming missions such as NISAR? Additionally, previous machine learning based studies that addresses sea ice classification needs to be addressed.

Method:
Section 2.1: Just curious, is there a reason by the original images were not used instead of screen captures? How does the resolutions compared? Could better results be expected if the original images were used?

Line 48/50: Is there a reference you can use where this approach was used before? I think the method of using RGB for classification used here needs to be backed by past studies or justified using patterns drawn from the images collected.

Section 2.2
Some basic information about the L4 data needs to be provided at the beginning to make it easier for the reader to follow along.

Figure 1, 2  The captions needs more details. The quality of figure needs to improve. There seems to be unnecessary white space. The size of the three images need to be made consistent. Please include what the white dots represent and is this relative to the ship in the center?

Line 64/65: Can you further explain what you mean by the leading edge effect here?

Line 85: Which issues are you referring to here? Be specific.

Line 90: Extra 'a' here..

Line 100: There needs to be a justification of why you are using this particular model for this classification task.

Line 117: Are there past studies that has made this observation about CNNs?

Line 130: What was the size of this ground truth dataset? How many images were used?

Figure 9: Needs improvements. Very difficult to undertsand this plot.

Table 1: Its not clear to me whether a part of the original dataset used for training and testing was used for validation or a separate dataset using the ground truth collocated photos were used to validate the model? Are these the results from the validation test using the collocated photos?

Section 3.3: This is actually a very interesting analysis. I feel like there could be further insights and interesting observations made from this which would further strengthen the study. I recommend adding more content to further justify the performance of the model compared to the salinity and temperature measurements.

---

## Author Comment (AC1)

We express our most sincere gratitude to the two anonymous referees who took the time to review our manuscript and provide edits as well as constructive feedback. We feel that all the recommendations acted to strengthen the manuscript and we were able to address and respond to each of the referees' comments. In the detailed response below we indicate how we modified the manuscript to address their comments and include relevant quoted revised text. The referees' comments are shown in *black italic text* and our response in blue with direct quotes from the revised manuscript indented.

**RC1: 'Comment on egusphere-2025-643', Anonymous Referee #1, 06 May 2025**

*This paper looks to classify sea ice locations in ship-based radar imagery using a deep learning convolutional neural network (CNN). This is a novel application of a CNN, and in general the paper presents a good experimental design. My main concern is that this manuscript is very short. In particular, it contains no discussion section to contextualise the findings of this paper to recent relevant literature. I also detail below some ways in which the methods section could be restructured, because some sections are very large and encompass multiple parts of the workflow. It would instead be easier for the reader if it was broken down.*

*Introduction: Nice and concise, but jumps too quickly from discussing ice detection in Passive microwave imagery to on-board radar. For context a lot of machine learning based research has been conducted using optical and SAR satellite imagery that also overcomes some of the limitations of the passive microwave derived sea ice concentration record. In the introduction and discussion, this project should be contextualised against this body of literature e.g.*

*de Gélis, I., Colin, A., Longépé, N., 2021. Prediction of Categorized Sea Ice Concentration From Sentinel-1 SAR Images Based on a Fully Convolutional Network. IEEE Journal of Selected Topics in Applied Earth Observations and Remote Sensing 14, 5831–5841. https://doi.org/10.1109/JSTARS.2021.3074068*

*Rogers, M.S.J., Fox, M., Fleming, A., van Zeeland, L., Wilkinson, J., Hosking, J.S., 2024. Sea ice detection using concurrent multispectral and synthetic aperture radar imagery. Remote Sensing of Environment 305, 114073. https://doi.org/10.1016/j.rse.2024.114073*

*Stokholm, A., Wulf, T., Kucik, A., Saldo, R., Buus-Hinkler, J., Hvidegaard, S.M., 2022. AI4SeaIce: Towards Solving Ambiguous SAR Textures in Convolutional Neural Networks for Automatic Sea Ice Concentration Charting. IEEE Transactions on Geoscience and Remote Sensing 1–1.* https://doi.org/10.1109/TGRS.2022.3149323

We thank reviewer #1 for their helpful comments and the references they provide. We have included mention of these SAR and satellite based papers in the introduction section:

"Direct observation of sea ice concentration is important because passive microwave satellite measurements struggle to quantify SIC during the melt season (e.g., Kern et al., 2020). Ship-based S-band radar has been shown to be useful in the identification of sea ice, although other wavelengths can prove to be better for sea-ice characterizations (Haykin et al., 1985). Lund et al. (2018) used a shipboard X-band marine navigation radar to detect and track sea ice. Ku-band radar was used to characterize the types of sea ice observed, including first-year ice, multilayer ice, icebergs, or the shadows cast by icebergs (Orlando et al.,1990). However, classifying individual sea-ice radar images is labor intensive. Previous studies have used satellite observations together with machine-learning methods to detect and classify ice using synthetic aperture radar (SAR) (de Gélis et al.,2021; Stokholm et al., 2022) along with multi-spectral imagery (Rogers et al., 2024). Building on these methods we develop a machine-learning method to characterize sea ice concentration images captured by a shipboard S-band radar that would otherwise be used for navigational purposes only."

*Section 2.1. Please provide a map of the rough locations of the expedition. I note Figure 9 provides information on ship tracks, but a larger scale image of the whole region would be useful for context.*

We have now included a map showing the whole ship track (Figure 10), which is colored by sea ice classification from the S-Band radar.

*Line 40- interesting to see that the S-band acquisition was paused during extensive periods out of the sea ice. Is there an estimate of the proportion of footage within the four categories (Level 0 -3) within the training data?*

To ensure that the model was trained on correct data we made sure to include the same number of images from the four classifications (C0-C3). We have added mention of this to the text:

"These issues with using S-band radar to classify ice led us to abandon our simple ratio estimation and develop a machine learning model to classify the images. To this end, a subset of 1,100 of the 24,000 L4 Geo-TIFF images were chosen in non-sequential order to remove near-duplicates to develop a training data set that were label using a simple 4-level classification with ice concentration ranging from a zero (C0) to three (C3). The training data set included roughly an equal number of images from each of the four classification levels. "

*Line 47-49: "Color values consistent with yellow to red color ranges were classified as "sea ice", and those consistent with blue and green colors were classified as "no data". I reocgnise that you refer to Druushka (2024) for more detail, but please provide a short summary as to why yellow-red colour ranges correspond to sea ice.*

Thank you for pointing out we neglected to explain this, which we now do in the text in question:

"Details on how the screen capture device was set up and the steps used to convert the images (Fig. 2a) into geolocated bitmaps (Fig. 2b) are described in detail in Drushka et al., 2024 and summarized below. The metadata was extracted from the images using MATLAB's computer vision toolbox. Metadata include the spatial range of the image, which varied in size from 1.4 km to 22 km. The pixels outside of this range were removed from each image, and pixels within the range were georectified and stored as Geo-TIFF images. The pixels were classified according to their red, green and blue (RGB) values. Color values consistent with yellow to red color ranges were classified as "sea ice", and those consistent with blue and green colors were classified as "no data" because that is the color used by the Furuno display to indicate range rings and the radar scanning line that were not retained for the analysis."

*Line 89- typo: estiuamte*

The typo has been fixed, thanks!

*Section 2.3. This section is a bit mixed and needs breaking into smaller subsections. A large proportion actually discusses the data selection and augmentation techniques rather than the ML model itself. Please separate the text describing and justifying the use of the VGG-19 architecture and modifications with the text on data selection and augmentation.*

We thank the reviewer for pointing this out and have broken the old section 2.3 into two sections and revised and recorded text in the new section 2.4:

"2.3 Developing a Classification Scheme and Training Dataset
We chose to use ice levels versus directly estimating ice coverage because our interest is to identify whether the ship was in ice, around ice, or not in ice. This 4-level classification has a value of C0 when there is no ice at all in the frame of the radar image and the ship is in open water (Fig 6a). Noteworthy variation in the L0 images was observed due to different amounts of noise present on the radar image due to deployed instruments (Wave Gliders, SWIFT buoys, etc.; see Drushka et al 2024). The C1 ice concentration includes images with just a few individual ice floes in the frame (Fig 6b). For radar captures where the ship was in low to medium concentrations of ice (~10% or more), or there is high concentration of ice in the frame but the ship is not passing through it, a C2 classification was used (Fig 6c). Finally, a C3 classification was used to indicate that the ship is passing through sea ice with concentration in excess of about 30% (Fig 6d). Higher sea ice concentrations were not encountered because the ship does not have ice breaking capabilities.

2.4 The Machine Learning Model
The machine learning (ML) model is well suited to use images that contain random anomalies like those from sea clutter in the S-Band radar images, and CNNs are particularly well-suited for tasks like this where visual patterns are perceptible to humans

but difficult to formalize into specific rules. Since CNNs can learn these insights directly from labeled data (e.g., human-labeled ice levels), the model can be trained without requiring hand-crafted rules or filters. This is ideal in situations where human experts can label examples but cannot precisely specify what distinguishes one level of ice from another in terms of pixel properties. CNNs bridge this gap by mapping visual patterns to labels directly, learning filters that capture various levels of detail—from edges and textures to larger structural patterns—enabling the network to distinguish between visual nuances like the difference between low-ice concentrations and sea-clutter, that resist more simplistic rules-based methods.

The machine learning (ML) model we used for image classification is an off-the-shelf implementation of the VGG-19 network in PyTorch (Smonyan and Zisserman, 2014). The original VGG-19 is designed for the ImageNET benchmark task, and thus we reduce the final fully connected layer from 100 to 4 dimensions, and leverage transfer-learning by initializing the model with pre-trained weights (sinski et al., 2014). Predictions are generated using the softmax activation function on the final fully connected layer (Bridle 1990). VGG-19 is a complex and computationally expensive architecture, and so risks overfitting and can require the use of multi-core GPUs on large datasets (Huang et al., 2017). Here, we are able to run the model to convergence on a local CPU in less than an hour, as the single-channel image dataset is just 26MB on disk. We prevent overfitting by regularizing the model through the use of dropout and data augmentation. Dropout in a CNN (Convolutional Neural Network) is a regularization technique that randomly deactivates a fraction of neurons during training to prevent overfitting and improve the network's ability to generalize to new data (Srivatava et al., 2014). Our best performing model used dropout probabilities of 0.6 on the two fully connected layers, such that 60% of the neurons in these layers are randomly set to zero for each training example. In addition to dropout, we regularized the model using a data augmentation protocol that took advantage of the rotational invariance of the images. The S-band radar observes a circular region around the vessel, and thus the assigned class should be invariant to any rotational changes to the input. Here, we used an augmentation protocol that rotated 75% of the images vertically and/or horizontally by up to 50%, such that just 12.5% of input images are fed to the model in their original orientation (as in Bridle 1990).The model was trained for 15 epochs using a batch size of 64, and a train/test split of 85/15 and 5-fold cross-validation. The model achieved 91% accuracy on the test data, with class-specific performance shown in Table \ref{tab:accu} and confusion matrix in Figure 7. The model shows high performance on all 4 classes, and does not misclassify any of the testing data by more than one ice level (ex. C0 are only misclassified as C1)."

*Line 86- 87: How were the 1100 images selected? There are a number of dimensions e.g. zoom level, proportion of ice in the image and location. How was it ensure that the training dataset was not biased and contained representation of the entire dataset.*

The reviewer points out that we might have created a biased training dataset. When creating the training data set, we ensured that it had an approximately even split of images from different zoom levels (ranges) and ship speeds. This information is included in the new section 2.3 starting at line XX:

>""2.3 Developing a Classification Scheme and Training Dataset
>We chose to use ice levels versus directly estimating ice coverage because our interest is to identify whether the ship was in ice, around ice, or not in ice. This 4-level classification has a value of C0 when there is no ice at all in the frame of the radar image and the ship is in open water (Fig 6a). Noteworthy variation in the L0 images was observed due to different amounts of noise present on the radar image due to deployed instruments (Wave Gliders, SWIFT buoys, etc.; see Drushka et al 2024). The C1 ice concentration includes images with just a few individual ice floes in the frame (Fig 6b). For radar captures where the ship was in low to medium concentrations of ice (~10% or more), or there is high concentration of ice in the frame but the ship is not passing through it, a C2 classification was used (Fig 6c). Finally, a C3 classification was used to indicate that the ship is passing through sea ice with concentration in excess of about 30% (Fig 6d). Higher sea ice concentrations were not encountered because the ship does not have ice breaking capabilities."

*Figure 2- please provide more details within this image to aid readers who have not worked with this type of imagery before: What is the spatial extent of the images, what are the scales of the signal and what do the different colours in each image represent.*

Since the images shown in Fig. 3 are from the training dataset, we edited the caption to read as follows (Line XXX). Note that Fig. 2 in the original manuscript is now Fig. 3 in the revised manuscript.

>"Example L4 Geo-TIFF images created during the S-Band radar data processing that show multiple examples of sea clutter. The range or geographic extent of these images is not reported in this figure as the images are sourced from the training dataset, from which the text showing range, orientation, ship speed, etc. has been removed as part of the processing. (a) Example L4 Geo-TIFF image with minor unidentified sea clutter, (b) major unidentified sea clutter, and (c) unidentified sea clutter while in the ice"

*Figure 3: Again as someone who has not interpreted this type of imagery before, it is difficult to see what point you are making. There are a number of differences between image a) and b), which parts of the image(s) show the effects of returns off of the leading edge of the ice pack? You make reference to northwest and southeast. Is North at the top of all images, please clarify.*

To help readers understand what we are showing in the figures we augmented Fig. 3 with lines showing the sea ice edge and arrows pointing to ice beyond the edge. We edited the figure caption as follows (Line XXX):

"Example L4 Geo-TIFF image from September 9 2022 at (a) 17:51 and (b) 17:54 showing the effect of returns off of the leading edge of the ice pack (red line in panel a) and returns off of ice not visible when first approaching the dense patch of ice (blue arrows in panel b). The range or geographic extent of these images is not reported in this figure as the images are sourced from the training dataset which did not include range, orientation or ship speed.

*Figure 4- Again what is the orientation and scale of these two images. Please add scale bars and north arrows to each image or clarify elsewhere.*

We thank the reviewers for their input on this and have modified the figure caption as follows:

"Example L4 Geo-TIFF images two minutes apart showing how changes in the radar setting by the ship's captain, in this case changes in radar range with (a) being zoomed out (larger range) in comparison to (b), affect the detection of sea ice. The range or geographic extent of these images is not reported in this figure as the images are sourced from the training dataset which did not include range, orientation or ship speed."

Please note that we did not include a North arrow as the information is not retained in the training data, nor is it relevant to the model.

*Figure 7: Do you have the corresponding radar images for these pictures again to aid the reader.*

We have now included a second column in Fig. 8 (previous Fig. 7) that includes the S-band radar images concurrent with the pictures.

*Figure 11- very nice figure- how did you choose the number of bins.*

In an attempt to make sure our span of data in both temp and salinity was fully represented in the figure we chose the bin size to convert the span of the data and still be small enough to show differences between bins.

**RC2: 'Comment on egusphere-2025-643', Anonymous Referee #2, 16 May 2025**

*Westbrook et al. (2025) propose a machine learning-based approach for classifying sea ice concentration using ship-board S-band radar imagery. This is a novel contribution with substantial scientific merit. The study is methodologically sound, particularly due to its use of multiple validation datasets, which significantly enhances its robustness. However, I recommend major revisions before the manuscript is ready for publication. One of the key shortcomings is the absence of a dedicated discussion section. The current results section does not engage in*

*analytical interpretation, and the conclusion is overly brief, leaving the manuscript feeling incomplete. More critical reflection on the implications, limitations, and broader context of the findings is necessary to fully realize the potential of this work.*

We thank the reviewers for their suggestions and comments and have worked to expand the discussion of interpretation of our results or how it fits into the body of work already established on this subject. The discussion and conclusions section have been heavily modified, starting at line (XXX). For clarity we did not paste these changes in this response and instead point to the track-changes version of the revised manuscript.

*Also, The quality of figures need to be improved before publication. The captions need to have more information. Some general comments that need to be addressed:*

As the other referee pointed out as well, more info was needed in the captions. We have edited the captions to include what is in these images. Furthermore we explain in the figure caption that these images are from the training dataset and do not retain information of zoom scale, direction of travel, or speed. Please see figure captions 2-7.

*Abstract: I feel the abstract needs to establish the uniqueness of the research as well as the ending sentence should put the applicability of this work in broader context.*

We thank referee #2 for their suggestions and modified the abstract as follows:

> "To gain context on the ambient sea ice field during the 2022 NASA Salinity and Stratification at the Sea Ice Edge (SASSIE) expedition we developed a machine learning model to predict sea ice cover classification from screen captures of a ship-board S-band navigation radar. The SASSIE expedition measured ocean surface properties and air-sea exchange approximately 400 km north of Alaska in the Beaufort Sea for 20 days, during which time screen captures from the shipboard S-band radar were collected. Our goal was to analyze these images to determine when the ship was approaching sea ice, in the ice, or in open water. Here we report on the development of a machine learning method built on the PyTorch software packages to classify the amount of sea ice observed in individual radar images on a scale from L0-L3, with L0 indicating open water and L3 assigned to images taken when the ship was navigating through thick sea ice in the marginal ice zone. The method described here is directly applicable to any radar images of sea ice and allows for the classification and validation of sea ice presence or absence. Furthermore, this method uses a standard marine navigation radar that is not generally used to measure sea ice and thus opens the opportunity to categorize sea ice concentration using the type of navigation radar installed on most vessels."

*Introduction: The introduction is too short and needs more details to set the context and uniqueness of the study. Particularly, focused should be given on why classification of S-band radar and any past studies that has looked into this? Also, how is this helpful from a satellite*

*remote sensing context, is this relevant for upcoming missions such as NISAR? Additionally, previous machine learning based studies that addresses sea ice classification needs to be addressed.*

We thank the reviewer for the suggestion to expand our introduction. We have included discussion of how this work fits into previous work using SAR. This material starts at line XXX:

"In the Arctic, the advance of the autumn sea ice has become slower and occurs later in the season, whereas the retreat of the summer ice has become faster and occurs earlier in the season, both as a direct result of climate change (Stroeve and Notz, 2018). As more of the ocean surface is exposed to the atmosphere during longer periods of open water, air-sea heat and momentum fluxes, the freshwater cycle, surface albedo feedback, primary production, and regional and global climate, as well as human and ecological health are changing (Lannuzel et al., 2020). The NASA Salinity and Stratification at the Sea Ice Edge (SASSIE) mission aims to study the role that surface salinity and stratification have on controlling the freezing of sea ice. The SASSIE expedition took place in the marginal ice zone during fall 2022 in the western Beaufort Sea around 72°–73.5°N, 154°–145°W (Drushka et al., 2024). This area was generally covered with sea ice year-round until the late 2000s, when the marginal ice zone moved northward during the summer melt season. Drastic changes in sea ice cover are occurring throughout the Arctic and direct observations of sea ice are needed to help understand and document the changes.

Direct observation of sea ice concentration is important because passive microwave satellite measurements struggle to quantify SIC during the melt season (e.g., Kern et al., 2020). Ship-based S-band radar has been shown to be useful in the identification of sea ice, although other wavelengths are typically better for sea-ice characterizations (Haykin et al., 1985). Lund et al. (2018) used a shipboard X-band marine navigation radar to detect and track sea ice. Ku-band radar was used to characterize the types of sea ice observed, including first-year ice, multilayer ice, icebergs, or the shadows cast by icebergs (Orlando et al., 1990). However, classifying individual sea-ice radar images is labor intensive. Previous studies have used satellite observations together with machine-learning methods to detect and classify ice using synthetic aperture radar (SAR) (de Gélis et al.,2021; Stokholm et al., 2022) along with multi-spectral imagery (Rogers et al., 2024). Building on these methods we develop a machine-learning method to characterize images captured by a shipboard S-band radar that would otherwise be used for navigational purposes only.

The classification of sea ice types from S-band radar images lends itself well to the use of machine learning models in part because the images are invariant to rotation. The ubiquity of machine-learning classification methods in Earth Science has resulted in open-source tools that can be implemented on diverse datasets without the need to develop complex software required to build such models. In this project we use the open-source PyTorch machine learning library https://pytorch.org/."

*Method:  Section 2.1: Just curious, is there a reason by the original images were not used instead of screen captures? How does the resolutions compared? ould better results be expected if the original images were used?*

In order to record the data we had to use a screen capture device as the ship-board system did not have a way to save the images.  The screen captures had the same resolution as what was displayed on the radar.

*Line 48/50: Is there a reference you can use where this approach was used before? I think the method of using RGB for classification used here needs to be backed by past studies or justified using patterns drawn from the images collected.*

We were not able to find any references that used the RGB values to represent binary ice/no ice.  We did update the methods to describe how the RGB scales to return strength:

> "... The pixels outside of this range were removed from each image, and pixels within the range were georectified and stored as Geo-TIFF images. The pixels were classified according to their red, green and blue (RGB) values. Color values consistent with yellow to red color ranges were classified as "sea ice", and those consistent with blue and green colors were classified as "no data" because that is the color used by the Furuno display to indicate range rings and radar scanning line that were not retained for the analysis."

*Section 2.2  Some basic information about the L4 data needs to be provided at the beginning to make it easier for the reader to follow along.  Figure 1, 2  The captions needs more details. The quality of figure needs to improve. There seems to be unnecessary white space. The size of the three images need to be made consistent. Please include what the white dots represent and is this relative to the ship in the center?*

All figures in question have been rearranged and the captions have been expanded.  An example is the caption below for Fig 2 (Fig. 1 in the original manuscript)

> "Example images recorded from the S-band radar display onboard the R/V Woldstad. The actual screen capture from the Furuno S-band radar display is shown in panel (a) and the accompanying L4 geolocated image used in this analysis in panel (b).  The ship is at the center of the images and sea clutter is visible around the ship in panel (a). North is oriented up in both panels.  The line clearly visible across the radar image is the current direction the radar is facing as it scans.  Sea ice is visible as yellow and red pixels in panel (a) that are then converted into a geolocated binary image shown in panel (b)."

Note that cations to figures 3 and 4 have been updated as well.

*Line 64/65: Can you further explain what you mean by the leading edge effect here?*

We have modified this section to expand on our discussion of the leading edge effect:

> "When the ship approaches a patch of sea ice, the amount of ice in the floe is underestimated due to the radar's detection of the leading edge of the ice, that is, the ice floes that are closest to the ship. The ship's radar bounced off of the leading ice floes creating an acoustic shadow, so ice floes in the shadow were not detected. This is revealed in two radar captures taken 3 minutes apart as the ship enters a dense patch of ice (Fig. 4). This leading edge of the ice floe is shown in red and in Fig 4b blue arrows indicate ice floes missing or underestimated because they were in the shadow of larger ice floes. The result of this leading edge effect is an underestimation of ice concentration in close proximity to the ship when simple ratios of black and white pixels are used to analyze these images. In the machine learning model described below we are able to account for this distortion by including numerous examples in the training data."

*Line 85: Which issues are you referring to here? Be specific.*

The sentence in question has been modified to detail which issues we faced:

> "These issues with using S-band radar to classify ice, including sea clutter, leading edge effects, and radar setting changes, led us to abandon our simple ratio estimation and develop a machine learning model to classify the images."

*Line 90: Extra 'a' here..*

Fixed, thanks!

*Line 100: There needs to be a justification of why you are using this particular model for this classification task.*

The justification for our choice of model was based on finding an off-the-shelf model that would be suitable for the task. We included this in the text:

> "The machine learning (ML) model we used for image classification is an off-the-shelf implementation of the VGG-19 network in PyTorch (Simonyan and Zisserman, 2014). The original VGG-19 is designed for the ImageNET benchmark task and we found it to be a good match to the problem we faced. To reduce the complexity of the model to suit our task we reduce the final fully connected layer from 100 to 4 dimensions, and initialize the model with pre-trained weights.

*Line 117: Are there past studies that has made this observation about CNNs?*

We found the following study that stated that CNNs are good for sea clutter identification

Enhanced CNN-based Small Target Detection in Sea Clutter with Controllable False Alarm" by Q. Qu et al., published in IEEE Sensors Journal.

This reference has been added to the text in question.

*Line 130: What was the size of this ground truth dataset? How many images were used?*

Details were added to the text:

> "Throughout the SASSIE expedition, notes documenting the state of the ice were recorded in the ship's log (N=31) and geolocated photos (N=10) were taken from the deck of the R/V Woldstad while the ship was in ice."

*Figure 9: Needs improvements. Very difficult to undertsand this plot.*

We expanded the caption of this figure to better explain what is shown.  We welcome any suggestions that could help make this figure easier to understand.

> "The number of measurements recorded from each ice classification in temperature and salinity bins as a 2D histogram.  Bar color indicates sea ice classification with the z-axis indicating the number of observations in a given temperature and salinity bin. Bin size was chosen to ensure the representation of the full dynamic range of the data and to  be able to differentiate between individual bins."

*Table 1: Its not clear to me whether a part of the original dataset used for training and testing was used for validation or a separate dataset using the ground truth collocated photos were used to validate the model? Are these the results from the validation test using the collocated photos?*

We tested the model initially using a random split of 85/15 training/test of the images.  This is explained in the text and now also in the caption of the figure 7:

> "Confusion matrix of the predictive performance of the final CNN model used in the analysis. The model accuracy was estimated by testing the model on a subset of 15% of the images that were randomly held out of the training dataset.  The class accuracy is shown along the diagonal. Notably, no examples in the testing data have been misclassified to an ice level that is more than 1 step away. For example, no C0 are classified into C2 or C3, no C2 are classified as C0, etc."

*Section 3.3: This is actually a very interesting analysis. I feel like there could be further insights and interesting observations made from this which would further strengthen the study. I recommend adding more content to further justify the performance of the model compared to the salinity and temperature measurements.*

We thank the referee for suggesting that we expand this section.  We have now included a new discussion and conclusions section that discusses the observed role surface salinity has in estimated sea-ice classification along with discussion of how the method can be used on existing data and suggestions for future efforts.

> "To classify sea-ice concentration into four categories during a 20-day expedition into the Beaufort Sea we employed an open-source machine learning model trained on a subset of manually classified S-band radar images.  Our model was successful at predicting a subset of training data between 84% and 95% of the time when tested on a random 85/15 split of the training data (Fig. 7). Using this model we classified over 24,000 images and compared these classifications to both photos taken from the ship as well as the Alaska Sea Ice Program gridded ice maps and shipboard measurements of temperature and salinity.  This simple model yielded predictions that fit into the temperature and salinity expected for differing ice coverage. It is important to note that the middle two classifications (C1 and C2) were difficult to define in the training data and also yielded complex relationships with temperature and salinity suggesting that our methods are best suited for determining if there is ice or not, versus the amount of ice present.  Overall, this method provides an example of using existing machine learning methods to create value-added data products from what generally is considered navigation radar and not traditionally used to characterize sea ice.

> The distribution of ice classification from the S-band radar images in temperature and salinity space (Fig. 12) highlight the influence surface salinity has on ice concentration with the two highest categories of sea-ice (C3-C4) are almost exclusively observed in water fresher than 25, regardless of temperature.  Higher sea-ice classifications were observed in warmer waters (up to 2 C) only at salinity less than 24.5 PSU.

> The methods developed here are applicable to any images or radar returns of sea ice. Existing radar and image observations of sea ice can use the methods developed here together with a curated training dataset to classify sea-ice concentrations. The application of methods developed here to existing ship-based radar images could provide important observational data of the marginal ice zone where validation data can be hard to find (Pacini et al., 2025). Future efforts to extract sea-ice concentration from S-band radar images would be improved if continuous photographs are taken from the platform hosting the radar. Video or time-lapse images would help to build a more accurate training dataset and thus increase model precision "